# Prevalence and Spatial Distribution of Animal Brucellosis in Central Punjab, Pakistan

**DOI:** 10.3390/ijerph17186903

**Published:** 2020-09-21

**Authors:** Usama Saeed, Shahzad Ali, Tooba Latif, Muhammad Rizwan, Anam Iftikhar, Syed Ghulam Mohayud Din Hashmi, Aman Ullah Khan, Iahtasham Khan, Falk Melzer, Hosny El-Adawy, Heinrich Neubauer

**Affiliations:** 1Wildlife Epidemiology and Molecular Microbiology Laboratory (One Health Research Group), Discipline of Zoology, Department of Wildlife & Ecology, University of Veterinary and Animal Sciences (UVAS), Lahore, Ravi Campus, Pattoki 55300, Pakistan; usamasaeeduvas@gmail.com (U.S.); tooba7latif@gmail.com (T.L.); rizwanasif400@gmail.com (M.R.); attaullah.saif@uvas.edu.pk (A.); ghulam.mohayudin@uvas.edu.pk (S.G.M.D.H.); 2Department of Biological Sciences, University of Veterinary and Animal Sciences, Lahore, Ravi Campus, Pattoki 55300, Pakistan; anam.iftikhar@uvas.edu.pk; 3Department of Pathobiology, College of Veterinary and Animal Sciences, Jhang 35200, Pakistan; Amanullah.khan@uvas.edu.pk; 4Section of Epidemiology and Public Health, Department of Clinical Sciences, University of Veterinary and Animal Sciences, Lahore Sub-Campus, Jhang 35200, Pakistan; Iahtasham.khan@uvas.edu.pk; 5Institute of Bacterial Infections and Zoonoses, Friedrich-Loeffler-Institut, Naumburger Str. 96a, 07743 Jena, Germany; Falk.Melzer@fli.de (F.M.); Hosny.ElAdawy@fli.de (H.E.-A.); Heinrich.Neubauer@fli.de (H.N.); 6Faculty of Veterinary Medicine, Kafrelsheikh University, Kafr El-Sheikh 33516, Egypt

**Keywords:** brucellosis, seroprevalence, Arc GIS, inverse distance weight (IDW)

## Abstract

Brucellosis is an important zoonotic disease of animals and humans caused by bacteria of the genus *Brucella*. Brucellae are Gram-negative intracellular bacteria which infect a wide variety of animals including goats, sheep, buffaloes, cows, pigs, and wildlife. The objectives of this study were to determine the seroprevalence and spatial distribution of brucellosis in Central Punjab, Pakistan. A total of 1083 blood samples of goats, sheep, buffaloes, and cows were collected from 38 villages of four districts (Kasur, Faisalabad, Lahore, and Okara) of Punjab, Pakistan, and screened for brucellosis by Rose Bengal Plate test (RBPT) and PCR confirmed. Epidemiological, demographic data and GPS coordinates for every sample were collected. By using interpolation of the Aeronautical Reconnaissance Coverage Geographic Information System (Arc GIS), a surface plot was generated applying inverse distance weight (IDW). It was found that 35 (3.23%) serum samples were positive for brucellosis. In eight (61.5%), six (75%), seven (87.5%), and eight (89%) villages, positive goats, sheep, buffaloes, and cattle were detected, respectively. In general, older animals are more often positive for brucellosis. In goats bucks were more often RBPT positive than females while in sheep, buffaloes, and cattle more females were positive. The spatial distribution of brucellosis shows that it is widely distributed in the western region of the study area in goats and in the South-West region in sheep. Similarly, for buffaloes it is restricted to the south-east and north-west regions, and in cattle brucellosis is present in western region of study area only. Reflected by this study, brucellosis poses a risk for livestock in developing countries due to lack of awareness by officials, owners, and consumers, and control measures are missing. A risk map of brucellosis was generated to develop effective strategies for awareness rising and to improve the quality of control programs in Pakistan.

## 1. Introduction

Livestock plays a pivotal role in the economy of Pakistan and it is considered the backbone of rural economy because more than 70% of the population lives in rural areas and incomes depend on animal production [1]. Livestock products contributed 58.92% to the gross domestic product (GDP) of Pakistan during the financial year 2017/18. Total growth of livestock production during this period was recoded to be 3.76%, which is a considerable increase when compared to that of the proceeding years [2]. Livestock production uplifted the socioeconomic status of resources poor farming communities in Pakistan.

There are ten agro-ecological zones in Pakistan categorized by climate, water availability of land, and land use, which may influence the spatial and temporal distribution of livestock diseases [3]. Brucellosis is one of the most important zoonotic diseases which was eradicated from developed countries but remained endemic in developing countries due to lack of resources for control programs [4]. Still birth, infertility in male animals, birth of weak calves, and abortion are the major symptoms of livestock brucellosis, which cause substantial economic loss to farmers [5].

Bacteria of the genus *Brucella* (*B.*) cause brucellosis. They are Gram-negative intracellular bacteria which infect a wide variety of animals including farm animals like goats, sheep, buffaloes, cattle, pigs, and the wildlife [6].

In small ruminants the causative agent of brucellosis is *Brucella melitensis*, which is most often also the cause of diseases in humans. *Brucella abortus* and *Brucella suis* are also recognized zoonotic agents. In bovines the major causative agent of brucellosis is *B. abortus* [7]. If bovines are kept together with goats and sheep or pigs, *B. melitensis* and *B. suis* interchange may occur. In different countries like Canada, Australia, and New Zealand *B. abortus* has been eradicated [8]. Brucellae are transmitted in animals by contact to vaginal discharge, placenta, and fetal fluids from the infected animals. Brucellae are also present in milk and semen [9].

There are various methods used for diagnosis including culture and phenotypic identification of brucellae and a variety of serological tests has been developed for easy and rapid diagnosis of brucellosis [10]. False positive results might be seen in serological tests so additionally techniques like PCR and LAMP were used on serum samples to improve sensitivity [11]. In addition to technical peculiarities such as incubation temperature of serological tests, many extrinsic factors like vaccination or endemic status can influence sensitivity and specificity of these tests also [12].

There are many risk factors for animal brucellosis in the fields of animal management and environment. Risk factors related to animals are age, sex, abortion history, parity, and milking method [5,13]. Risk factors related to management are awareness of disease, hygiene, vaccination, breeding practice, and herd size [14]. A risk factor related to the environment is the agro-ecological location of the farm [15].

There is large variety of symptoms related to brucellosis in bovines and small ruminants, but abortion is the most striking symptom [16]. Repeated insemination, retention of the placenta, reduction of milk production, orchitis, and metritis are other common clinical signs [17]. Using different serological tests, the overall prevalence of brucellosis in bovines of different areas of Pakistan was determined to be 6.5% [18,19].

Despite the impact of brucellosis for public health of Pakistan, there is no procedure implemented to make data available for public health authorities to rapidly control outbreaks. The spatial modeling of diseases like brucellosis using the capability of geographical information systems is appropriate to recognize the spatial variation of diseases and its relationship with epidemiological, demographic, and other factors. Geographical information systems are widely used as effective tools in public health of developed countries, but their application is very restricted in developing countries like Pakistan. The present study was aimed to determine seroprevalence and association with spatial distribution of animal brucellosis at village level in selective districts of Central Punjab, Pakistan.

## 2. Materials and Methods

### 2.1. Study Area

The cross sectional study was conducted in four different districts, Lahore (31.5204° N, 74.1350° E), Kasur (31.1165° N, 74.4494° E), Okara (30.8090° N, 73.4508° E), and Faisalabad (30.14504° N, 73.1350° E) of the Punjab, Pakistan, from July 2017 to July 2018. A samples size of 74 herds was estimated based on following parameters: herd seroprevalence (*p*) of 5.01% [20], desired absolute precision (*d*) of 0.05 and 95% confidence interval using the method *n* = (1.96)^2^
*p*(1 − *p*)/*d^2^* [21]. Two herds were randomly examined from each village (38 villages) due to the absence of complete herd record. Blood samples of 50% of the animals from each herd were taken for best representation due to wide variations in herd sizes. These districts are rich of indigenous flora and fauna. Farm animal breeds kept in these areas are Nili Ravi buffaloes, Sahiwal cattle, Beetal and Teddy goats, and Kajli sheep. Lahore is linked with India by the Wagah border. In the districts Kasur Okara and Faisalabad livestock is produced in intensive and semi intensive systems. More prevalent diseases of livestock in the study area are mastitis, foot and mouth disease, milk fever, and simplex and herpes virus infection.

### 2.2. Arc GIS Based Survey

The Arc GIS based survey was conducted from October 2017 to March 2018 through GPS [(Global positing system (Michael Schollmeyer, Seattle, WA, USA)] receiver (Application installed in Mobile phone version 4.4.25). Geographical coordinates were acquired from each sampling site.

### 2.3. Epidemiological Data

Demographic related information (district = 4, and village = 38) and characteristics of animals (i.e., species and gender) were collected on the sampling day by using a questionnaire.

### 2.4. Blood Sampling and Serum Preparation

A total of 1083 blood samples were collected from sheep, goats, buffaloes, and cattle. About 4 mL blood was taken from jugular vein with a disposable sterile syringe and transferred to a tube with clot activating factor (non EDTA tubes). The non EDTA sample tubes were kept at room temperature for one hour and transported on ice to UVAS, Lahore, Ravi Campus, Pattoki. Serum was separated by centrifugation at 6000 rpm for 4 min and sera were kept at −20 °C for further study.

### 2.5. Serological Investigation Using Rose Bengal Plate Test (RBPT)

The RBPT was performed according to manufacturer’s instructions (ID. Vet, France). Accordingly, 30 µL serum was placed on a glass plate and the equal quantity of antigen was added and then mixed gently. After mixing the glass plate was agitated for 4 min and any agglutination was considered positive [22]. Control sera were provided by Institute of Bacterial Infections and Zoonoses, Friedrich-Loeffler-Institute, Jena, Germany.

### 2.6. Extraction of DNA and Real-Time (RT)-PCR

Thirty five RBPT positive serum samples were further investigated for the presence of *Brucella* DNA using genus specific RT-PCR. DNA extraction was done using the PCR template preparation kit as per company’s instructions (Roche Diagnostic, Mannheim, Germany). Concentration and purity of DNA were checked using ND-1000 UV visible spectrophotometer (Nano-Drop Technology, Wilmington, DE, USA). The primers and probes were supplied by TIB MOLBIOL, Berlin, Germany. The reactions were performed in duplicate using a MX3000P PCR Machine (Applied Biosystem, Darmstadt, Germany). *B. melitensis* 16M (ATCC23456) and *B. abortus* S-99 (ATCC23448) DNA were used as positive controls while nuclease free water was used as negative control. An internal amplification control (IAC) was used. The RT-PCR was performed as originally described [23].

### 2.7. Development of Geographic Information System (GIS) Maps

Spatial distribution was investigated by generation of GIS map using Arc GIS 10.5.1 software package (Michael Schollmeyer, Seattle, WA, USA). Inverse distance weight (IDW) was used to investigate spatial trends.

### 2.8. Statistical Analysis

Statistical analysis was done using Statistical Package for Social Sciences (SPSS) version 21.0 software (IBM Armonk, New York, USA) and association between risk factors was analyzed by Fisher’s Exact Test. Multiple logistic regression was used to find out the 95% confidence interval levels, odd ratios and association between each risk factors and prevalence of brucellosis. A *p* value ≤ 0.05 was considered statistically significant [24].

## 3. Results

Out of 1083 serum samples, 35 (3.23%) samples were positive for brucellosis in RBPT and were confirmed by PCR.

Thirteen and eight villages were investigated for caprine and ovine brucellosis and eight (61.5%) and six (75%) villages were found positive, respectively. For brucellosis prevalence in buffaloes and cattle, eight and nine villages were examined and seven (87.5%) and eight (89%) were found to be positive, respectively.

The impact of different factors was investigated on the geographical distribution of animal brucellosis. In buffaloes, cattle and sheep, female animals were more likely to be infected as males. However, more males (3.4%) were seropositive in goats (*p* ≤ 0.05) (Table 1).

The risk for infection was greater in buffaloes and cattle older than 4 years i.e., 10.8% and 9%, respectively. In sheep and goats older than 3 years, the prevalence was 8.9% and 3.9%, respectively (*p* ≤ 0.05) (Table 2).

However, gender and age were identified as potential risk factors for brucellosis in buffaloes, cattle, and goats based on multiple logistic regression analysis (Table 3). Sheep of 1.6–3 years were at high risk of acquiring brucellosis.

The highest prevalence in goats were found in the villages Kot Saring Wala, district Kasur (3.2%) and Renala Khurd, district Okara (3.2%) while in Manga Mandi, district Lahore it was 0% and in Dijkot district Faisalabad 3.1%. The highest prevalence of brucellosis in sheep was detected in Renala Khurd village, district Okara (7.1%), (Table 4).

The highest prevalence of brucellosis in buffaloes (9.1%) was observed in the village Chak 64 JB, district Faisalabad when compared to villages of other districts, and in cattle prevalence of 7.1% were found in Dijkot Faisalabad, 5.2% in Makkuana, 4.7% in Chak 64 JB, and 4.5% in Jagu Wala Chak 40, district Faisalabad (Table 4).

By using the method of interpolation in Arc GIS with the help of IDW (inverse distance weight) a surface was generated (Table 4). The spatial trends in goats of the study area were shown by the following map. The seroprevalence was recorded highest in the western part of the area (Figure 1A). The prevalence there was 2.889–3.099%. Lowest prevalence were recorded in the North and South shown in dark green color. Similarly, a map of prevalence for sheep was generated. The highest prevalence (3.669–6.792%) were found in the South West and lowest (0.373–3.497%) in the North East (Figure 1B).

Prevalence in buffaloes is shown in Figure 1. Highest prevalence were found (5.411–9.222%) in the South East and North West (red color). The central part of the study area is shown in light green, orange and dark green in the South represent the lowest prevalence (Figure 1C).

Figure 1D shows the prevalence in cattle. Faisalabad Saddar region had the highest prevalence (6.667–7.043%) followed by Chak 64 JB and Makkuana regions of Faisalabad and Lahore, respectively. Kasur and Okara regions had lowest prevalence mirrored by dark green color (0.00–2.234%).

The results of present study were compared with earlier studies accompanied in different areas of Punjab Pakistan described in (Table 5). Variation has been seen in prevalence of brucellosis in various animal host tested from various localities of Pakistan.

## 4. Discussion

Brucellosis causes significant economic losses by abortion, loss in fertility, milk production, and costs for replacement of animals. It is considered the second most destructive zoonotic disease of the world after rabies. Most of the cases are seen in underdeveloped countries.

In the present study RBPT/PCR prevalence of 3.23% was found. The serological results were confirmed by PCR as these authors are aware of the shortcomings of RBPT. This prevalence is much lower than that observed in a recent study of the Potohar plateau region of Pakistan (8.6%) [24]. The possible reason of this difference might be the different geographical location. A lower prevalence was observed (2.7%) in a previous study by [34] in Bangladesh. Persistence of disease is caused by the farmers who are not disposing positive animals but sell them to other farmers [24].

A variation has been seen in the village wise prevalence of brucellosis in goats (61.5%), sheep (75%), buffaloes (87.5%), and cattle (89%) in the present study. Variations in cattle and buffaloes at village level were reported from Punjab Pakistan earlier [35]. The present prevalence is much higher than that found by [36] in Bardsir province Kerman, Iran, where 6% prevalence was found in goats and sheep and 4.3% in cattle. The village-wise prevalence of brucellosis was higher in our study. The probable reason for this could be the husbandry system in these villages. The animals of these villages are kept in small herds and mixing of multiple species is common practice. Moreover, the villagers use common grazing grounds and cohabitation/grazing of farm animals species. They get rid of obviously diseased and not reproductive animals because they lose money, but animals with sufficient production stay in the herds. They never become visible or ignored as the loss is acceptable.

The highest prevalence was observed in female buffaloes. This result is agreement with results reported from the district Peshawar of Pakistan by [37]. Quite opposite results were described by [26,27], from different sites of Pakistan with higher prevalence in males (7.4% and 12%) and lower ones in females (2.5 and 2.89%), respectively. The possible reason might be that positive males are not replaced by the farmers and breeding is a continuous source of transmission of disease.

Data for bucks (3.4%) and for female (0.8%) goats are in contrast to earlier results of [29,38], who found female herds often seropositive. The present results are also opposite to results of a study conducted in Michoacan, Mexico, in small ruminants with 9% prevalence in females and 5% in males [39]. Again, it can be speculated that failure to replace the male animals or selling them to other farmers are source of chronic infection of the herds. Data for female sheep are in agreement with the results of a study conducted in Rawat, Islamabad and Kherimurat Pakistan by [27].

In cattle the highest prevalence was seen in cows (7%), a situation which is comparable to that in Nyagatare district, Eastern province, Rwanda [40], and in different regions of Egypt [41]. The possible reason is use of the same management system of rearing that conserves the transfer of brucellosis to the next generation [42].

*Brucella* antibodies were more often prevalent in old animals. This is true for all four species investigated and in agreement with results of studies in small and large ruminants conducted in four different districts of Punjab Pakistan (3.3%) by [24], and in Ethiopia (8.47%) by [43]. In previous studies conducted in various countries (Ethiopia and Pakistan) the same trend was also observed [16,26,44]. The possible reason is that brucellosis is a disease of sexually mature animals and growth and multiplication of *Brucella* may be promoted by presence of sex hormones and erythritol in mature animals [45]. It can also be assumed that old animals had more often contact to infected animals and so they are more likely to be positive.

Geospatial techniques are the state-of-the-art techniques in epidemiology to identify the spatial trends of a disease. The high prevalence of brucellosis in goats of the villages of the district Kasur might be explained by different grazing systems [46].

Ovine brucellosis is more prevalent in the villages Manga Mandi, Renala Khurd, and Dijkot of districts Lahore, Okara, and Faisalabad, respectively. A previous study in the same districts found lower prevalence [24]. However, comparable data for villages of Iran have been published with 6% of the villages seropositive [36]. Most probable reason is that a different study design was used in both studies to describe the trend on map. The finding of few animals positive in the herd but with a considerable number of herds being affected points to the fact that the disease is chronic for a long time in the area.

The prevalence in buffaloes (Figure 1C) was high in the villages Chak 64 JB district Faisalabad, and Manga mandi district Lahore, and Chunian district Kasur. Hence, it was much lower when compared to a previous study conducted on the Potohar Plateau, Pakistan by [25]. The possible reasons for this controversy are different landscapes, environmental conditions, and diagnostic techniques. The trend on map clearly shows that buffaloes are affected by brucellosis. The most probable reason of this result is that buffaloes are the common diary animals in study area. Farmers cannot identify diseases of animals and if they are forced to keep sick animals due to economic needs [46].

Figure 1D shows the brucellosis trend in cattle. Although the disease was not found in Kasur, Lahore, and Okara villages, the map shows that brucellosis is a common disease of cattle. Dijkot village of the district Faisalabad showed the highest prevalence. One of the possible reasons for this finding might be that the main cattle market of Pakistan is located in this district. Moreover, the multispecies farming system is common in this region with cattle and buffaloes as dominant species; although, a few individual sheep and goats are reared for private use. Thus, presence of various animal species on a farm favor the chance of cross species infection. A higher prevalence was observed in a previous study conducted in indigenous and Holstein–Friesian cattle breed farms of the Sind Province, Pakistan, by [47]. A lower prevalence (1.1%) was found in a previous study of West Algeria by [48].

As zoonotic and infectious disease which effects human as well as animal’s health, the management of brucellosis at farm level is a highly demanding and challenging task. *Brucella* species and biotypes can exhibit sectorial geographic distribution even in restricted geographical areas. This observation is most often associated with poor herd management procedures as *Brucella* infection is favored when healthy and infected animals are mixed. Thus, there is dire need for the implementation of biosafety and biosecurity measures at farm level. Indeed, animal exchange or contacts between different farms must be controlled e.g., via pre-entry testing and quarantine to prevent intra- and inter-herd pathogen spreading [49,50,51,52]. Moreover, vaccination against brucellosis in farm animals is the demand of time. To ensure sustainability the characteristics of the landscape and the poor living conditions of farmer families have to be the guiding aspects of any strategy to be developed.

## 5. Conclusions

For surveillance and to strengthen disease control programs knowledge on seroprevalence and spatial trends of diseases are important. For better implementation of vaccination and other control strategies in different geographical regions of Pakistan this information helps to prioritize funding. Affected animals are source of disease. Replacement of sero-reactors and development of effective control policies are demands for the moment in Pakistan.

## Figures and Tables

**Figure 1 ijerph-17-06903-f001:**
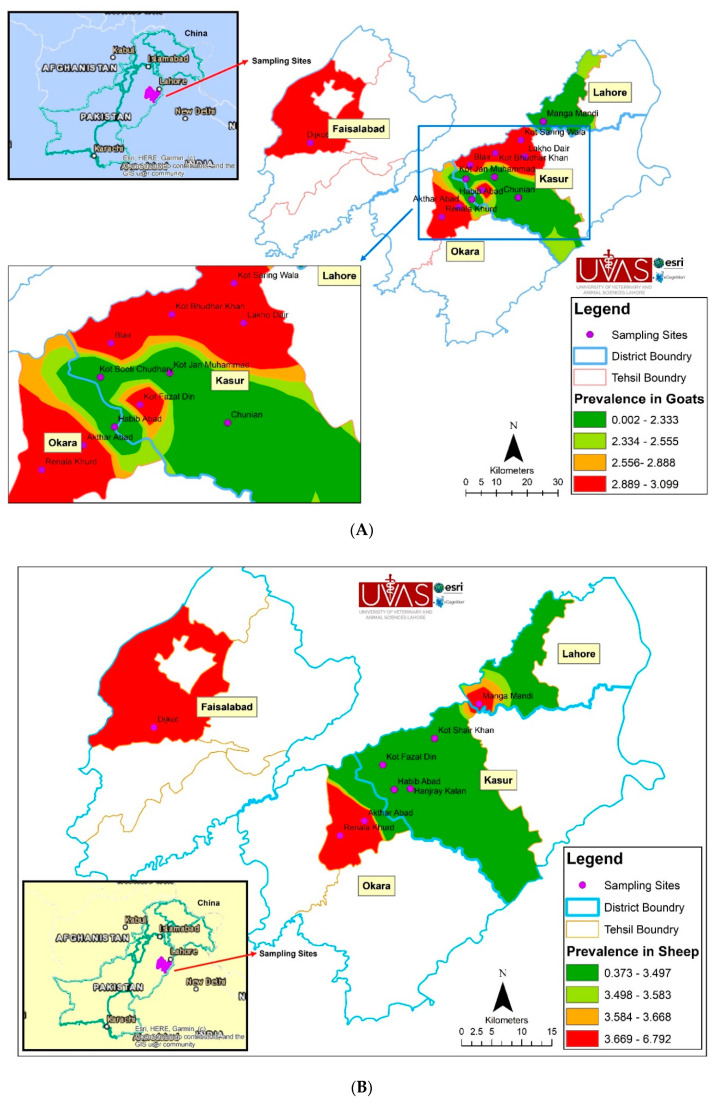
(**A**) Spatial Trends of Brucellosis in Goats; (**B**) Spatial Trends of Brucellosis in Sheep; (**C**) Spatial Trends of Brucellosis in Buffaloes; (**D**) and Spatial Trends of Brucellosis in Cattle in Punjab, Pakistan.

**Table 1 ijerph-17-06903-t001:** The effect of species and sex on seroprevalence of brucellosis.

Species	Gender	Examined	Positive	Prevalence %	Fisher Exact Test (*p* Value)
Buffaloes	Male	122	3	2.5	0.05
	Female	112	9	8	
Goats	Male	179	6	3.4	0.05
	Female	261	2	0.8	
Sheep	Male	105	1	1	0.04
	Female	98	6	6.1	
Cattle	Male	120	2	1.7	0.05
	Female	86	6	7	

**Table 2 ijerph-17-06903-t002:** The effect of species and age on the prevalence of brucellosis in Punjab, Pakistan.

Species	Age	Examined	Positive	Prevalence %	Fisher Exact Test (*p* Value)
Buffaloes	≤2 Years	85	2	2.3	0.04
	2–4 Years	74	2	2.7	
	≥4 Years	75	8	10.8	
Cattles	≤2 Years	60	0	0	0.01
	2–4 Years	68	1	1.5	
	≥4 Years	78	7	9.0	
Sheep	≤1.5 Years	78	1	1.3	0.05
	1.6–3 Years	69	1	1.4	
	≥3 Years	56	5	8.9	
Goats	≤1.5 Years	168	3	1.8	0.04
	1.6–3 Years	144	0	0	
	≥3 Years	128	5	3.9	

**Table 3 ijerph-17-06903-t003:** Association of gender and age of buffaloes, cattle, goats, and sheep with *Brucella* prevalence based on Multiple Logistic Regression Analysis for the Punjab, Pakistan.

Species	Variable	Factors	Odd Ratio	*p* Value	95% CI
Buffalo	Gender	Male	0.138	0.007	0.033–0.578
		Female	Ref.		
	Age	≤2 Years	0.099	0.006	0.019–0.522
		2–4 Years	0.127	0.015	0.024–0.665
		≥4 Years	Ref.		
Cattle	Gender	Male	0.114	0.013	0.020–0.633
		Female	Ref.		
	Age	≤2 Years	−20.324	0.001	1.490–009
		2–4 Years	0.059	0.012	0.007–534
		≥4 Years	Ref.		
Goat	Gender	Male	93.193	0	9.850–881.74
		Female	Ref.		
	Age	≤1.5 Years	0.019	0	0.002–0.157
		1.6–3 Years	−23.941	0	4.005–011
		≥3 Years	Ref.		
Sheep	Gender	Male	0.101	0.08	0.008–1.337
		Female	Ref.		
	Age	≤1.5 Years	0.515	0.61	0.038–7.05
		1.6–3 Years	0.102	0.04	0.011–0.919
		≥3 Years	Ref.		

**Table 4 ijerph-17-06903-t004:** The effect of host and village in different districts on the prevalence of brucellosis in Punjab, Pakistan.

Host	Districts	Tehsil	Name of Village	Number Examined	Number Positive	Seroprevalence (%)
Goats	Kasur	Pattoki	Kot Bhudhar Khan	41	1	2.4
			Kot Saring Wala	31	1	3.2
			Kot Booti Chudhary	31	0	0.0
			Kot Fazal Din	34	1	2.9
			Kot Jan Muhammad	35	0	0.0
			Lakho Dair	32	1	3.1
			Blair	36	1	2.7
			Habib Abad	35	0	0.0
		Chunian	Chunian	34	0	0.0
	Okara	Renala Khurd	Renala Khurd	31	1	3.2
			Akthar Abad	34	1	2.9
	Lahore	Lahore	Manga Mandi	34	0	0.0
	Faisalabad	Faisalabad	Dijkot	32	1	3.1
Sheep	Kasur	Pattoki	Kot Shair Khan	29	1	3.4
			Hanjray Kalan	24	0	0.0
			Kot Fazal Din	29	1	3.4
			Habib Abad	27	0	0.0
	Okara	Renala Khurd	Renala Khurd	28	2	7.1
			Akthar Abad	24	1	4.1
	Lahore	Lahore	Manga Mandi	26	1	3.8
	Faisalabad	Faisalabad	Dijkot	16	1	6.2
Buffaloes	Kasur	Pattoki	Munday Kay	27	1	3.7
		Chunian	Chunian	28	2	7.1
	Okara	Renala Khurd	Renala Khurd	27	0	0.0
			Akthar Abad	33	2	6
	Lahore	Lahore	Manga Mandi	32	2	6.3
	Faisalabad	Faisalabad	Dijkot	23	1	4.4
			Chak 64 JB	33	2	9.1
			Makkuana	31	2	6.5
Cattle	Kasur	Pattoki	Lakho Dair	25	1	4
			Shakim	24	1	4.2
			Buruj Mahalum	20	0	0.0
			Jagu Wala Chak 40	22	1	4.5
	Okara	Renala Khurd	Akthar Abad	25	1	4
	Lahore	Lahore	Manga Mandi	26	1	3.8
	Faisalabad	Faisalabad	Dijkot	24	1	7.1
			Chak 64 JB	21	1	4.7
			Makkuana	19	1	5.2

**Table 5 ijerph-17-06903-t005:** Comparison of present and earlier studies conducted on brucellosis in different areas of Pakistan.

References	Region	Species	Prevalence	Diagnostic Method
[25]	Potohar Plateau	Bovines	6.3% by RBPT, out of 170 positive samples, 52.4% positive for qRT-PCR, 6.7% positive in MRT.	RBPT, MRT, culturing and qRT-PCR
[26]	Potohar Plateau	Bovines	6.9% (Cattle) and 6.6% (Buffalo).	MRT
[27]	Potohar Plateau including Rawat, Islamabad and Kherimurat	Small Ruminants	8.6% by RBPT, 9.4% by MRT and out of 24 positive samples 18 (75%) by qRT-PCR.	RBPT, MRT and qRT-PCR
[24]	Kasur, Okara, Faisalabad and Lahore districts	Bovines and Ovines	3.23% by RBPT and qRT-PCR.	RBPT and qRT-PCR
[28]	Mirpur and Azad Kashmir	Bovines and Ovines	8.6% by RBPT and 6.87% by ELISA.	RBPT and ELISA
[29]	Pattoki and Karachi regions	Birds, and Wild Animals	11.1% by RBPT.	RBPT
[20]	Potohar Plateau	Cattle	5.01% by RBPT, 4.76% by SAT, and 3.25% by qRT-PCR.	RBPT, SAT, qRT-PCR
[30]	Faisalabad and Bahawalpur	Canines and Livestock	37.6% by SAT in Dogs, 4.9% in livestock by ELISA, and 1% by PCR.	SAT, ELISA, and PCR
[31]	Jhang, Chiniot and Bhakkar	Camel	5% by RBPT, 2% by CELISA, and 1.5% by PCR.	RBPT, CELISA and PCR
[32]	Hyderabad district	Cattles and Buffaloes	31.88% and 47.19% observed in cattle and buffalo, respectively, by MRT.	MRT
[33]	Faisalabad	Horses	20.7% by RBPT and 17.7% by SAT.	RBPT and SAT

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
