# Peer review of "Prevalence and Spatial Distribution of Animal Brucellosis in Central Punjab, Pakistan"

_ijerph, 2020, doi:10.3390/ijerph17186903_

Round 1
Reviewer 1 Report
The study describes the seroprevalence and spatial distribution of brucellosis in Central Punjab, Pakistan. For this purpose, a total of 1,083 blood samples collected from goats, sheep, buffaloes and cows bred in four districts (Kasur, Faisalabad, Lahore and Okara) of Punjab were screened for brucellosis by Rose Bengal Plate test. Epidemiological, demographic data and GPS coordinates were analyzed to generate a surface plot by using the Aeronautical Reconnaissance Coverage Geographic Information System (Arc GIS), and applying Inverse distance weight (IDW). Results display different levels of seroprevalence related to different variables, such as animal species, age, sex and geographical location of the farms.
This study clearly shows the presence of a serological response to brucellosis in the area of interest, and the risk maps presented in the manuscript allow a simple and direct view of the presented data. The paper is not particularly original but it can be useful in the veterinary field for the set-up and application of effective control programs. However, the manuscript needs to be implemented and several issues need to be addressed.
First, the manuscript needs careful language editing and should be corrected by a native English speaker.
Is the number of animals sampled for each farm representative of the total number of animals present in that farm?
The authors perform the RBPT methods for identification of the presence of antibodies specific for Brucella spp. Do the authors have any chance to perform a Complement Fixation Test? No single serological test is appropriate in all epidemiological situations and all animal species; all tests have limitations especially when screening individual animals. CFT is more specific than the RBPT, and has also a standardized system for unitage (OIE).
Presented data seem to indicate that there is a clear difference between the seroprevalence levels exhibited by the Faisalabad region in comparison with the other regions for both sheep and cattle. Can the authors provide a possible explanation for this result? Do the authors have any information on possible factors responsible for such clear spatial variation of seroprevalence levels?
Authors should provide information about the time period of samples collection.
Lines 183-184: the sentence “A variation has been seen in the village wise prevalence of brucellosis in goats (61.5%), sheep (75%), buffaloes (87.5%) and cattle (89%)” is not clear to me. I did not understand if this is a data reported by the authors in this work or cited from published reports.
Do the authors have any chance to provide information about the abortion occurrence in the tested farms? It would be a useful information to be added to the manuscript.
Also, can the authors collect any information on the milk yield production in the interested farms?
Biosafety measures are essential for animal health at farm level, and the authors should highlight this concept in their manuscript. Brucella species and biotypes can exhibit sectorial geographic distribution even in restricted geographical areas. This evidence is mostly associated with efficient herd management procedures. Indeed animal exchange or contacts between different farms must be avoided to prevent intra- and inter-herd pathogen spreading (see suggested references).
[Suggested references]
OIE. 2009. Manual of diagnostic tests and vaccines for terrestrial animals, chapter 2.4.3. OIE (World Organization for Animal Health), Paris, France.
Aznar MN, Arregui M, Humblet MF, Samartino LE, Saegerman C. Methodology for the assessment of brucellosis management practices and its vaccination campaign: example in two Argentine districts. BMC Vet Res. 2017 Sep 7;13(1):281.
Borriello G., Peletto S., Lucibelli M.G., Acutis P.G., Ercolini D., Galiero G. “Link between geographical origin and occurrence of Brucella abortus biovars in cow and water buffalo herds.”. Applied and Environmental Microbiology, 2013; 79(3): 1039-43.
Zhang Q, Li C, Wang Y, Li Y, Han X, Zhang H, Wang D, Liao Y, Chen Z. Temporal and Spatial Distribution Trends of Human Brucellosis in Liaoning Province, China. Transbound Emerg Dis . 2020 Jul 21.
Author Response
REVIEWER 1
Correction highlighted in yellow color text
Comments and Suggestions for Authors
The study describes the seroprevalence and spatial distribution of brucellosis in Central Punjab, Pakistan. For this purpose, a total of 1,083 blood samples collected from goats, sheep, buffaloes and cows bred in four districts (Kasur, Faisalabad, Lahore and Okara) of Punjab were screened for brucellosis by Rose Bengal Plate test. Epidemiological, demographic data and GPS coordinates were analyzed to generate a surface plot by using the Aeronautical Reconnaissance Coverage Geographic Information System (Arc GIS), and applying Inverse distance weight (IDW). Results display different levels of seroprevalence related to different variables, such as animal species, age, sex and geographical location of the farms.
This study clearly shows the presence of a serological response to brucellosis in the area of interest, and the risk maps presented in the manuscript allow a simple and direct view of the presented data. The paper is not particularly original but it can be useful in the veterinary field for the set-up and application of effective control programs. However, the manuscript needs to be implemented and several issues need to be addressed.
Comment 1: First, the manuscript needs careful language editing and should be corrected by a native English speaker.
Response: The manuscript was carefully corrected with the help of a native speaker.
Comment 2: Is the number of animals sampled for each farm representative of the total number of animals present in that farm?
Response: Blood samples were collected from 50% animals from each farm.
Comment 3: The authors perform the RBPT methods for identification of the presence of antibodies specific for Brucella spp. Do the authors have any chance to perform a Complement Fixation Test? No single serological test is appropriate in all epidemiological situations and all animal species; all tests have limitations especially when screening individual animals. CFT is more specific than the RBPT, and has also a standardized system for unitage (OIE).
Response: We have performed additionally genus specific RT-PCR [high sensitivity (95%) and specificity (100%)] for further confirmation of seropositive serum samples (Line 118-126). Only sera with a positive PCR result were used for further calculations. We are aware of the fact that the true prevalence could be higher. We have explained this procedure now in the manuscript.
Comment 4: Presented data seem to indicate that there is a clear difference between the seroprevalence levels exhibited by the Faisalabad region in comparison with the other regions for both sheep and cattle. Can the authors provide a possible explanation for this result? Do the authors have any information on possible factors responsible for such clear spatial variation of seroprevalence levels?
Response: The reason for higher prevalences of brucellosis in cattle/sheep of Faisalabad is explained better now (line 253-256).
Comment 5: Authors should provide information about the time period of samples collection.
Response: Time period is mentioned (July 2017 to July 2018) now, line 91.
Comment 6: Lines 183-184: the sentence “A variation has been seen in the village wise prevalence of brucellosis in goats (61.5%), sheep (75%), buffaloes (87.5%) and cattle (89%)” is not clear to me. I did not understand if this is a data reported by the authors in this work or cited from published reports.
Response: The sentence describes the data of the present study. The sentence is revised line 206.
Comment 7: Do the authors have any chance to provide information about the abortion occurrence in the tested farms? It would be a useful information to be added to the manuscript.
Response: We are grateful to your kind suggestions. However, we did not collected such data.
Comment 8: Also, can the authors collect any information on the milk yield production in the interested farms?
Response: We are grateful to your kind suggestions. However, we did not collected such data.
Comment 9: Biosafety measures are essential for animal health at farm level, and the authors should highlight this concept in their manuscript. Brucella species and biotypes can exhibit sectorial geographic distribution even in restricted geographical areas. This evidence is mostly associated with efficient herd management procedures. Indeed animal exchange or contacts between different farms must be avoided to prevent intra- and inter-herd pathogen spreading (see suggested references).
Response: The concept of measures to control brucellosis is now explained in detail (line 260-268).
OIE. 2009. Manual of diagnostic tests and vaccines for terrestrial animals, chapter 2.4.3. OIE (World Organization for Animal Health), Paris, France.
Aznar MN, Arregui M, Humblet MF, Samartino LE, Saegerman C. Methodology for the assessment of brucellosis management practices and its vaccination campaign: example in two Argentine districts. BMC Vet Res. 2017 Sep 7;13(1):281.
Borriello G., Peletto S., Lucibelli M.G., Acutis P.G., Ercolini D., Galiero G. “Link between geographical origin and occurrence of Brucella abortus biovars in cow and water buffalo herds.”. Applied and Environmental Microbiology, 2013; 79(3): 1039-43.
Zhang Q, Li C, Wang Y, Li Y, Han X, Zhang H, Wang D, Liao Y, Chen Z. Temporal and Spatial Distribution Trends of Human Brucellosis in Liaoning Province, China. Transbound Emerg Dis . 2020 Jul 21.
Reviewer 2 Report
line 87-94> I was expecting to see how come about the selection of the villages. You need to specify the type of epidemiological study and sampling method used for this study.
For your result in Table 2 and 3, you can not use Chi-square test as the important requirement for this Chi-Square test of all values in the cells must be greater than 5. In the majority of the positive cases were less than 5, so I expect you to use Fischer exact test for your analysis. I will suggest you redo this to assess for statistical significance.
While you may have done Chi square tests, I think you should try multivariable logistic regression. This allow you to test for the dependent variables on the seroprevalence. i.e. effects of location of sample, age, sex, and species on the brucella seroprevalence.
Author Response
REVIEWER 2
Corrections highlighted in red color text
Comments and Suggestions for Authors
Comment 1: line 87-94> I was expecting to see how come about the selection of the villages. You need to specify the type of epidemiological study and sampling method used for this study.
Response: The procedure of selection of villages, study type and sampling method are described now (line 91-95).
Comment 2: For your result in Table 2 and 3, you cannot use Chi-square test as the important requirement for this Chi-Square test of all values in the cells must be greater than 5. In the majority of the positive cases were less than 5, so I expect you to use Fischer exact test for your analysis. I will suggest you redo this to assess for statistical significance.
Response: We have reanalyzed data with Fisher exact test, line 132-134. Table 1 and 2 have been revised. Some p values changed slightly but did not affects significance of results.
Comment 3: While you may have done Chi square tests, I think you should try multivariable logistic regression. This allow you to test for the dependent variables on the seroprevalence. i.e. effects of location of sample, age, sex, and species on the brucella seroprevalence.
Response: The possible association of gender and age of buffaloes, cattle, goats and sheep for Brucella prevalence based on Multiple Logistic Regression Analysis has been described in Table 3 and line 150-154.

Reviewer 3 Report
In this study the authors report the results of a multistage serological survey for brucellosis in Central Punjab, Pakistan, in which 1,083 blood samples from domestic ruminants were tested using the Rose Bengal plate test (RBPT). Only 35 samples tested positive, giving a low seroprevalence. My main issue with the paper is that the known imperfect specificity of the RBPT is not taken into account. RBPT is mainly used as a screening test, and failure to use a more specific confirmatory serological test means that it is impossible to know whether some or all of the positives were false positives. The fact that in most “positive” villages only one seropositive animal was detected suggests that some or many of them may have been false positives.
The issue of false positives and imperfect sensitivity and specificity is mentioned by the authors in the Introduction (LL66-69). However, this issue is then disregarded when interpreting their own results. The results are also compared with those of previous studies without taking into account the differences between the types of tests used.
The design of the survey is inadequately described. How was the sample size calculated? What sampling frame(s) were used? How were villages/herds/animals selected? All this information must be given, including the exact type of random (or otherwise) selection methods that were used.
After collection, the blood samples were “immediately” stored on ice. The authors should explain why this was done, since it is usual to allow the blood to clot at room temperature for a period of time before chilling. I am not sure whether this may have affected the results of the serological test.
L120: It is not the “association between risk factors” that should be analyzed, but the association of each potential risk factor with the outcome. However, it is not mentioned what risk factors were analyzed, and the only factors that were mentioned were district, village, species and gender.
Due to the low observed seroprevalence many of the cell counts in the cross-tabulations are low, meaning that the chi-square test is not valid and a Fisher’s exact test should have been used.
Only univariate associations were tested. However, there may have been confounding between age, sex and other variables. If possible, multivariable analyses should be done to control for this.
Some of the tables are unnecessary because all the results are given in the text (Table 1) or because the detail is unnecessary (Table 4).
The maps should have inset maps to show the location within the wider region, for readers unfamiliar with Pakistan.
Author Response
REVIEWER 3
Correction highlighted in in green color text
Comments and Suggestions for Authors
Comment 1: In this study the authors report the results of a multistage serological survey for brucellosis in Central Punjab, Pakistan, in which 1,083 blood samples from domestic ruminants were tested using the Rose Bengal plate test (RBPT). Only 35 samples tested positive, giving a low seroprevalence. My main issue with the paper is that the known imperfect specificity of the RBPT is not taken into account. RBPT is mainly used as a screening test, and failure to use a more specific confirmatory serological test means that it is impossible to know whether some or all of the positives were false positives. The fact that in most “positive” villages only one seropositive animal was detected suggests that some or many of them may have been false positives.
Response: We have performed additionally genus specific RT-PCR [high sensitivity (95%) and specificity (100%)] for further confirmation of seropositive serum samples, now (Lines 119-126). All seropositive (n=35) animal were positive in RT-PCR. This is described in the manuscript clearly now.
Comment 2: The issue of false positives and imperfect sensitivity and specificity is mentioned by the authors in the Introduction (LL66-69). However, this issue is then disregarded when interpreting their own results. The results are also compared with those of previous studies without taking into account the differences between the types of tests used.
Response: We only used data of PCR positive sera so that it is clear now that RBPT was only used to identify the sera to be tested with PCR. This was done to safe money of course. However, we are aware of the problem that the true prevalence might be higher but we think for showing the trends we want to show the procedure is appropriate. This procedure is explained now more in detail and the text was corrected accordingly where it was necessary. Line 241-243
Comment 3: The design of the survey is inadequately described. How was the sample size calculated? What sampling frame(s) were used? How were villages/herds/animals selected? All this information must be given, including the exact type of random (or otherwise) selection methods that were used.
Response: The procedure of selection of villages, study type and sampling method is described now lines 91-95).
Comment 4: After collection, the blood samples were “immediately” stored on ice. The authors should explain why this was done, since it is usual to allow the blood to clot at room temperature for a period of time before chilling. I am not sure whether this may have affected the results of the serological test.
Response: Description of procedure was corrected now (lines 110-111).
Comment 4: L120: It is not the “association between risk factors” that should be analyzed, but the association of each potential risk factor with the outcome. However, it is not mentioned what risk factors were analyzed, and the only factors that were mentioned were district, village, species and gender.
Response: This point was raised also by reviewer one and we have included new text to explain the outcomes. Please see answer to reviewer 1. [The possible association of gender and age of buffaloes, cattle, goats and sheep for Brucella prevalence based on Multiple Logistic Regression Analysis has been described in Table 3 and line 150-154.]
Comment 5: Due to the low observed seroprevalence many of the cell counts in the cross-tabulations are low, meaning that the chi-square test is not valid and a Fisher’s exact test should have been used.
Response: We have reanalyzed data with Fisher exact test, line 132-134. Table 1 and 2 have been revised. Some p values changed slightly but did not affects significance of results.
Comment 6: Only univariate associations were tested. However, there may have been confounding between age, sex and other variables. If possible, multivariable analyses should be done to control for this.
Response: The possible association of gender and age of buffaloes, cattle, goats and sheep for Brucella prevalence based on Multiple Logistic Regression Analysis has been described in Table 3 and line 150-154.
Comment 7: Some of the tables are unnecessary because all the results are given in the text (Table 1) or because the detail is unnecessary (Table 4).
Response: Table 1 was deleted. Moreover GPS coordinates and repetition in names of districts and tehsil were removed from Table 4.
Comment 8: The maps should have inset maps to show the location within the wider region, for readers unfamiliar with Pakistan.
Response: All maps were modified as suggested.

Round 2
Reviewer 1 Report
Authors adequately replied to all my comments
This manuscript is a resubmission of an earlier submission. The following is a list of the peer review reports and author responses from that submission.